# Inhaled anesthesia associated with reduced mortality in patients with stage III breast cancer: A population-based study

Emily Tzu-Jung Kuo[1,3]☯*, Chin Kuo[2,3]☯, Cheng-Li Lin[4,5]

1 Department of Anesthesiology, China Medical University Hospital, Taichung, Taiwan (R.O.C.), 2 College of Artificial Intelligence, National Yang-Ming Chiao Tung University, Tainan, Taiwan (R.O.C.), 3 School of Medicine, Duke University, Durham, NC, United States of America, 4 College of Medicine, China Medical University, Taichung, Taiwan (R.O.C.), 5 Management Office for Health Data, China Medical University Hospital, Taichung, Taiwan (R.O.C)

☯ These authors contributed equally to this work.
* tzu-jung.kuo@duke.edu

**Data Availability Statement:** Our study uses data from Taiwan's National Health Insurance Research Database (NHIRD), access to which is restricted due to legal and ethical regulations. For access inquiries, please contact the Health and Welfare

## Abstract

### Background

Patients diagnosed with stage III breast cancer often undergo surgery, radiation therapy, and chemotherapy as part of their treatment. The choice of anesthesia technique during surgery has been a subject of interest due to its potential association with immune changes and prognosis. In this study, we aimed to compare the mortality rates between stage III breast cancer patients undergoing surgery with propofol-based intravenous general anesthesia and those receiving inhaled anesthetics.

### Methods

Using data from Taiwan's National Health Insurance Research Database and Taiwan Cancer Registry, we identified a cohort of 10,896 stage III breast cancer patients. Among them, 1,506 received propofol-based intravenous anesthetic maintenance, while 9,390 received inhaled anesthetic maintenance. To ensure comparability between the two groups, we performed propensity-score matching.

### Results

Our findings revealed a significantly lower mortality rate in patients who received inhaled anesthetics compared to those who received propofol-based intravenous anesthesia. Sensitivity analysis further confirmed the robustness of our results.

### Conclusions

This study suggests that inhaled anesthesia technique is associated with a lower mortality rate in clinical stage III breast cancer. Further research is needed to validate and expand upon these results.

Data Science Center, Ministry of Health and Welfare: Ms. Zhong: +886-2-8590-6836, sta1229nita@mohw.gov.tw Ms. Huang: +886-2-8590-6821, stwenting@mohw.gov.tw Ms. Li: +886-2-8590-6809, stmelodyli@mohw.gov.tw.

**Funding:** This study was sponsored by China Medical University Hospital. The funders had no role in study design, data collection and analysis, decision to publish, or preparation of the manuscript.

**Competing interests:** The authors have declared that no competing interests exist.

## Introduction

Breast cancer is the most common diagnosed cancer globally and the second-leading cause of cancer-related death in the United States. Data sourced from The Global Cancer Observatory, an initiative focused on collating and disseminating comprehensive cancer research, reveals a global cumulative incidence rate of 5.20% for breast cancer [1]. Distressingly, an estimated 685,000 women lost their lives to breast cancer in 2020 [2]. Most patients received surgical interventions, making it crucial to explore the impact of perioperative factors on patient prognosis. Consequently, growing interests in understanding the role of perioperative factors in treatment outcomes and long-term survival for breast cancer patients.

The choice of anesthetic agents can impact both the host immune response and the progression of minimal residual disease in breast cancer. Despite the optimal treatment, minimal residual disease, characterized by circulating tumor cells (CTCs) and disseminated tumor cells (DTCs), continues to pose significant challenges due to subsequent local relapse and distant metastasis [3]. Generally, studies conducted on inhaled anesthetics have revealed their immunosuppressive and pro-inflammatory effects, as well as the potential to promote angiogenesis and cellular proliferation, facilitating the spread of cancer cells in various in vivo, in vitro, and animal models [4, 5]. On the other hand, propofol, commonly used in total intravenous anesthesia (TIVA), has been suggested to possess anti-inflammatory, antioxidative, and antitumor properties by directly regulating key pathways and signaling in cancer cells [6, 7]. However, the existing literature yields inconsistent findings, with some recent studies proposing a protective role for volatile agents [8]. Consequently, the effects of anesthetic agents on the progression of breast cancer remain incompletely understood, and conflicting evidence persists in preclinical research.

Breast cancer tumors were traditionally considered as "immune quiescence," with limited lymphocyte infiltration, low mutational burden, and modest response rates to anti-PD-1/PD-L1 monotherapy [9]. However, recent tumor and immunologic profiling has revealed potential mechanisms of immune evasion in breast cancer and unique aspects of the tumor microenvironment (TME) [9–12]. Crosstalk within the TME involving the extracellular matrix (ECM), vasculature, stromal cells, immune cells, and endothelial cells undergoes changes as the tumor progresses or in response to specific treatments [12]. Studies have demonstrated alterations in host immunity, including dysfunction and decreased numbers of Natural Killer (NK) cells associated with clinical stage [13, 14]. Additionally, adaptive immunity responds differently in advanced stages, with increased numbers of Regulatory T cells (Tregs) observed in the peripheral blood of breast cancer patients, correlating with invasive breast cancer [15, 16]. Given the differential host immune responses between early and advanced stages, we hypothesize that the impact of inhaled anesthetics and propofol-based intravenous anesthesia would vary in locally advanced breast cancer. However, the majority of the clinical studies have predominantly focused on early-stage breast cancer, specifically stage I and II [17–19]. Thus, this study aims to investigate potential differences in mortality rates between stage III breast cancer patients who undergo surgery with propofol-based intravenous general anesthesia and those who receive inhaled anesthesia. The overview of the topic discussed in this article is presented in Fig 1.

## Materials and methods

### Database

Data were collected from Taiwan's National Health Insurance Research Database (NHIRD) and linked with the Taiwan Cancer Registry (TCR). Taiwan has a single-payer healthcare

**Curative Treatment of Stage III Breast Cancer:**
Surgery + Chemotherapy + Radiotherapy

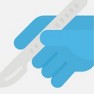 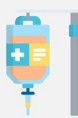 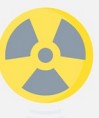

Regardless of the type of surgery performed, general anesthesia techniques can be categorized as propofol-based maintenance or inhaled maintenance.
The impact of general anesthesia techniques on prognosis remains unclear.

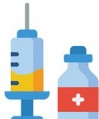 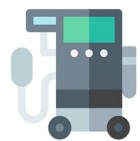

Propofol-based maintainance          Inhaled maintainance

**Fig 1. Topic overview.** Anesthesia techniques for curative surgery in stage III breast cancer can be categorized into two types: propofol-based maintenance or inhaled agent maintenance. This article aims to explore and compare the outcomes associated with these two techniques. Icon made by Freepik from www.flaticon.com.

system known as the National Health Insurance (NHI) Program, which has provided health insurance coverage to over 99.9% of the population since 1995 [20]. The NHIRD, derived from the NHI program, contains detailed information on medical claims, orders, expenses, prescriptions, and diagnoses for both inpatient and outpatient care. The Taiwan Cancer Registry, established in 1979, has high data quality and registered completeness of up to 97% [21]. We conducted a retrospective cohort study based on the population using de-identified data and accessed the database between August 15, 2021, and January 31, 2022, for research purposes. Ethical approval for the study was obtained from the Research Ethics Committee of China Medical University Hospital (IRB number: CMUH110-REC2–063), and informed consent was waived.

## Study population

We identified a cohort of patients who were initially diagnosed with clinical stage III primary breast cancer in the Taiwan Cancer Registry (TCR) using International Classification of Diseases codes (ICD-9-CM code: 174.9, ICD-10-CM code: C500-C509) and stratified them based on comorbidities and tumor risk factors. A total of 10,896 patients were selected. All the patients received standard treatment as recommended by the American Joint Committee on Cancer (AJCC) 7th edition guidance at the healthcare institute where they were diagnosed between 2010 and 2017. The eligible patients were divided into two groups: one group received inhaled anesthesia during surgery, and the other received propofol-based intravenous anesthesia. The inhaled anesthesia maintenance group was defined as receiving general anesthesia with less than 200mg propofol. The intravenous anesthesia group was defined as receiving total intravenous general anesthesia or general anesthesia with more than 200mg of propofol.

All of the patient cohorts were followed up for at least 2 years in the database, with the last follow-up date set on December 31, 2019. We excluded patients with a history of previous malignancy, patients with double cancer, and patients aged younger than 20 years.

## Outcome

In our study, we identified two research outcomes. The primary outcome was the mortality rate, emphasizing the overall mortality rate, as well as the 3-year and 5-year mortality rates. The overall mortality rate was calculated by dividing the number of deaths by every 1000 person-years in the at-risk population during the specified follow-up period. Our secondary outcome pertained to the overall recurrence rate, which was determined by dividing the number of recurrences by every 1000 person-years in the same high-risk population. Of note, the secondary endpoint may not be accurate due to the limitations of the database. We have discussed these limitations in the discussion section. Overall, our methodology facilitated a thorough assessment of both mortality and tumor recurrence as pivotal study endpoints.

## Statistical analysis

Demographic characteristics of patients with stage III breast cancer in the inhaled anesthetic and intravenous anesthetic groups were compared using t-tests for continuous variables and chi-square tests for categorical variables. Multivariate Cox proportional hazards regression models were used to derive adjusted hazard ratios (aHRs) and 95% confidence intervals (CIs) in each cohort, while adjusting for age, sex, comorbidities, and medications. The aforementioned factors are confounders that affect survival according to the literature [22, 23]. Propensity score matching was applied to reduce the impact of confounding factors. We generated cumulative mortality curves to describe the mortality rate over time in the intravenous group and inhaled group. All statistical analyses were performed using SAS System for Windows statistical software, version 9.4 (SAS Institute Inc., Cary, NC). The statistical significance criterion was set at a p-value of less than 0.05 for two-sided testing.

## Results

A total of 135,547 patients with breast lesions were identified from a merged database comprising NHIRD and TCR databases. Of these cases, 43,841 were excluded due to double cancer, prior cancer history, pathological non-malignancy, or non-breast malignancy histology. In Fig 2, we identified a cohort of 10,896 patients who had stage III breast cancer and had undergone surgical interventions.

Table 1 showed that of 10,896 patients with stage III breast cancer who received breast cancer surgery and standard treatment in Taiwan, 1,506 received propofol-based intravenous anesthetic maintenance, and 9,390 received inhaled anesthetics maintenance between 2010 and 2017. The mean age of diagnosis was 55.5 ± 12.2 years, and the median follow-up period was 4.71 years (interquartile range, 3.01–7.03) in the total study cohort. The patients were older and had more comorbid conditions, demonstrated as higher Charlson Comorbidity Index (CCI) in the intravenous group before the propensity score matching. Around 4.3% of patients who received more than 200mg of propofol in the inhalation-exposed groups may be caused by inaccurate medical claims or other medical conditions. Both the intravenous group and inhaled group demonstrate similar distribution on body mass index (BMI), CCI, 7th AJCC pathological stage, histological type, histologic grade, subtype, type of surgery, treatment sequence, received standard of care and year of diagnosis after 1:1 randomized propensity-matched. In the propensity-matched cohort, the patients exposed to the inhaled agent group

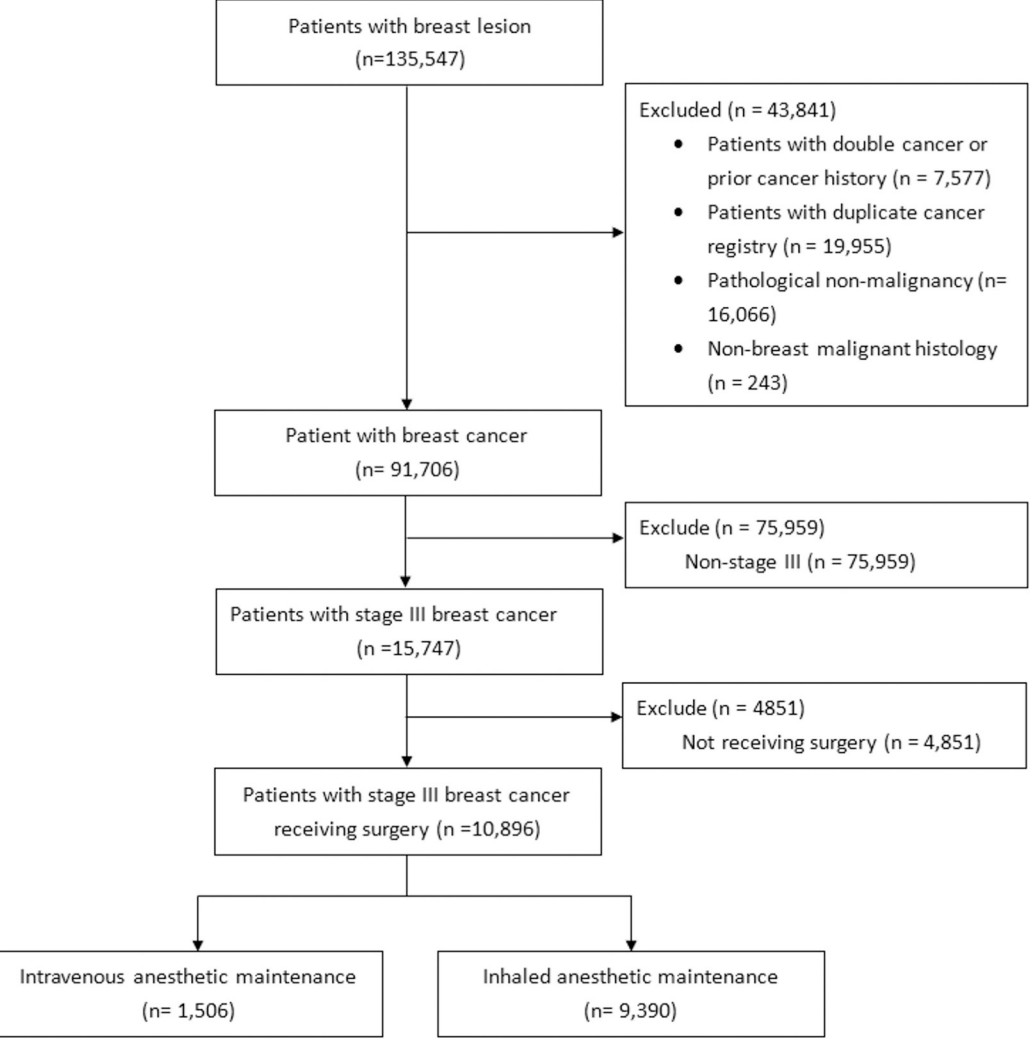

**Fig 2. Flowchart of study population.** The study analyzed patients with stage III breast cancer who underwent surgical interventions from NHIRD and TCR databases after excluding ineligible cases.

had longer median follow-up time, 6.09 years (interquartile range, 3.96–7.72), compared with the intravenous group, 3.66 years (interquartile range, 2.54–5.36).

Before propensity score matching, the overall mortality rate for stage III breast cancer patients who received maintenance with intravenous anesthetics is 6.39%, while the rate for those with inhaled anesthetic maintenance is 4.38%. After adjusting for age, sex, CCI, and medications, the patients who received inhaled anesthetics maintenance remained a lower overall mortality rate than those who received IV (Table 2, adjusted hazard ratio = 0.82, 95% CI: 0.72–0.93). We also found that patients aged above 50 at diagnosis, with comorbidities, normal BMI, 7th AJCC anatomic pathological stage IIIC, invasive ductal carcinoma, higher histologic grade, HER2-enriched subtype, and patients who received mastectomy were statistically significantly associated with a lower overall mortality rate, regardless of the anesthetic techniques. Furthermore, a subgroup analysis comparing IA to non-IA based on age revealed that the overall mortality rate was lower and statistically significant in the IA group for patients aged over 50 years.

**Table 1. Demographic characteristics of stage III breast cancer patients who received surgery in Taiwan (2010–2017).**

| Characteristic | Total study cohort, n(%) | | | p-value | Propensity matched cohort, n(%) | | | p-value |
|---|---|---|---|---|---|---|---|---|
| | Total | IV | IA | | Total | IV | IA | |
| | (n = 10896) | (n = 1506) | (n = 9390) | | (n = 2534) | (n = 1267) | (n = 1267) | |
| Age, year | | | | <0.001 | | | | 0.02 |
| 20–49 | 3587(32.9) | 432(12.0) | 3156(88.0) | | 730(28.8) | 381(52.2) | 349(47.8) | |
| 50–64 | 4932(45.3) | 633(12.8) | 4299(87.2) | | 1124(44.4) | 528(47.0) | 596(53.0) | |
| ≥65 | 2377(21.8) | 442(29.4) | 1935(20.6) | | 680(26.8) | 358(28.3) | 322(25.4) | |
| Mean (SD) | 55.5(12.2) | 57.7(13.2) | 55.2(12.0) | <0.001 | 57.1(12.8) | 57.2(13.2) | 57.0(12.3) | 0.65 |
| BMI, Mean (SD) | 24.8(4.48) | 25.5(4.73) | 24.7(4.42) | <0.001 | 25.4(4.67) | 25.4(4.72) | 25.4(4.62) | 0.88 |
| CCI | | | | <0.001 | | | | 0.17 |
| 0 | 7652(70.2) | 667(44.3) | 6985(74.4) | | 1279(50.5) | 632(49.9) | 647(51.1) | |
| 1–3 | 2321(21.3) | 493(32.7) | 1828(19.5) | | 808(31.9) | 424(33.5) | 384(30.3) | |
| >3 | 923(8.47) | 346(23.0) | 577(6.14) | | 447(17.6) | 211(16.7) | 236(18.6) | |
| pT category | | | | <0.001 | | | | 0.002 |
| T1 | 2012(18.5) | 278(18.5) | 1734(18.5) | | 495(19.5) | 238(18.8) | 257(20.3) | |
| T2 | 5112(46.9) | 662(44.0) | 4450(47.4) | | 1184(46.7) | 564(44.5) | 620(48.9) | |
| T3 | 1902(17.5) | 285(18.9) | 1617(17.2) | | 435(17.2) | 250(19.7) | 185(14.6) | |
| T4 | 968(8.88) | 112(7.44) | 856(9.12) | | 202(7.97) | 94(7.42) | 108(8.52) | |
| Unknown | 902(8.28) | 169(11.2) | 733(7.81) | | 218(8.60) | 121(9.55) | 97(7.66) | |
| pN category | | | | <0.001 | | | | 0.01 |
| N0 | 1559(14.3) | 182(12.1) | 1377(14.7) | | 345(13.6) | 152(12.0) | 193(15.2) | |
| N1 | 1654(15.2) | 213(14.1) | 1441(15.4) | | 354(14.0) | 181(14.3) | 173(13.7) | |
| N2 | 4355(40.0) | 585(38.8) | 3770(40.2) | | 1015(40.1) | 507(40.0) | 508(40.1) | |
| N3 | 2945(27.0) | 423(28.1) | 2522(26.9) | | 718(28.3) | 362(28.6) | 356(28.1) | |
| Unknown | 383(3.52) | 103(6.84) | 280(2.98) | | 102(4.03) | 65(5.13) | 37(2.92) | |
| Histology | | | | 0.11 | | | | 0.69 |
| IDC | 9379(86.1) | 1283(85.2) | 8096(86.2) | | 2167(85.5) | 1081(85.3) | 1086(85.7) | |
| ILC | 606(5.56) | 101(6.71) | 505(5.38) | | 162(6.39) | 86(6.79) | 76(6.00) | |
| Others | 911(8.36) | 122(8.10) | 789(8.40) | | 205(8.09) | 100(7.89) | 105(8.29) | |
| Grade | | | | 0.89 | | | | 0.08 |
| Gr.1 | 622(5.71) | 81(5.38) | 541(5.76) | | 140(5.52) | 58(4.58) | 82(6.47) | |
| Gr.2 | 5090(46.7) | 717(47.6) | 4373(46.6) | | 1199(47.3) | 609(48.1) | 590(46.6) | |
| Gr.3 | 4142(38.0) | 563(37.4) | 3579(38.1) | | 978(38.6) | 481(38.0) | 497(39.2) | |
| Unknown | 1042(9.56) | 145(9.63) | 897(9.55) | | 217(8.56) | 119(9.39) | 98(7.73) | |
| LVSI | | | | | | | | |
| Positive | 2563(23.5) | 360(23.9) | 2203(23.5) | <0.001 | 743(29.3) | 323(25.5) | 420(33.2) | 0.001 |
| Negative | 5451(50.0) | 857(56.9) | 4594(48.9) | | 1364(53.8) | 771(60.9) | 593(46.8) | |
| Unknown | 2882(26.5) | 289(19.2) | 2593(27.6) | | 427(16.9) | 173(13.7) | 254(20.1) | |
| Subtype | | | | | | | | |
| Luminal | 6621(60.8) | 976(64.8) | 5645(60.1) | <0.001 | 1706(67.3) | 852(67.3) | 854(67.4) | 0.93 |
| HER2 enrich | 1234(11.3) | 195(13.0) | 1039(11.1) | 0.03 | 366(14.4) | 179(14.1) | 187(14.8) | 0.65 |
| Basal | 965(8.86) | 140(9.30) | 825(8.79) | 0.52 | 242(9.55) | 125(9.87) | 117(9.23) | 0.59 |
| Unknown | 2076(19.1) | 195(13.0) | 1881(20.0) | | 220(8.68) | 111(8.76) | 109(8.60) | 0.89 |
| Type of surgery | | | | <0.001 | | | | 0.07 |
| BCS | 2104(19.3) | 275(18.3) | 1829(19.5) | | 472(18.6) | 248(19.6) | 224(17.7) | |
| Mastectomy | 7574(69.5) | 990(65.7) | 6584(70.1) | | 1775(70.1) | 862(68.0) | 913(72.1) | |
| Unknown | 1218(11.2) | 241(16.0) | 977(10.4) | | 287(11.3) | 157(12.4) | 130(10.3) | |
| Year of diagnosis | | | | <0.001 | | | | 0.91 |

(*Continued*)

**Table 1.** (Continued)

| Characteristic | Total study cohort, n(%) | | | p-value | Propensity matched cohort, n(%) | | | p-value |
|---|---|---|---|---|---|---|---|---|
| | Total | IV | IA | | Total | IV | IA | |
| | (n = 10896) | (n = 1506) | (n = 9390) | | (n = 2534) | (n = 1267) | (n = 1267) | |
| 2010–2012 | 4324(39.7) | 914(60.7) | 3410(36.3) | | 1437(56.7) | 723(57.1) | 714(56.4) | |
| 2013–2015 | 3867(35.5) | 480(31.9) | 3387(36.1) | | 888(35.0) | 442(34.9) | 446(35.2) | |
| 2016–2018 | 2705(24.8) | 112(7.44) | 2593(27.6) | | 209(8.25) | 102(8.05) | 107(8.45) | |
| Median follow-up (IQR) | 4.71 (3.01–7.03) | 3.60 (2.53–5.44) | 4.94 (3.12–7.23) | <0.001 | 4.64 (2.89–7.07) | 3.66 (2.54–5.36) | 6.09 (3.96–7.72) | 0.001 |

Abbreviation: IV, intravenous anesthetic group; IA, inhaled anesthetic group; BMI, body mass index; CCI, Charlson Comorbidity Index; IDC, invasive ductal carcinoma; ILC, invasive lobular carcinoma; LVSI, lymph-vascular space invasion; HER2, human epidermal growth factor receptor 2

We conducted a sensitivity analysis to assess the robustness of our hypothesis. Given the marked discrepancy in cohort sizes, with the intravenous (IV) cohort being significantly smaller than the inhaled cohort, we employed a 1:1 randomized propensity-matched approach to equalize the two groups. The mortality rates per 1,000 person-years were 57.9% for the IV cohort and 47.2% for the inhaled cohort, thereby reinforcing the validity of our empirical findings. After adjusting for potential confounders that could affect survival outcomes, the inhaled anesthetic cohort consistently showed a substantially lower overall mortality rate. This is supported by an adjusted hazard ratio (aHR) of 0.83 (95% CI: 0.71–0.98) compared to the IV cohort, as detailed in Table 3.

The cumulative mortality rate across the overall population is depicted in Fig 3. At each time point, the IV group's cumulative mortality rate consistently exceeded that of the inhaled group ($p < 0.001$). In the IV group, the mortality rate was 16.25% at 3 years and 26.17% at 5 years. In contrast, in the inhaled anesthesia (IA) group, the 3-year mortality rate was 13.2%, and the 5-year rate was 21.59% after propensity score matching. These findings indicate that stage III breast cancer patients who received inhaled anesthetics during surgery had better overall survival over time.

In Table 3, the inhaled cohort exhibited a higher recurrence rate, with an adjusted Hazard Ratio (aHR) for recurrence of 1.28 (95% Confidence Interval [CI]: 1.04–1.58) compared to the cohort receiving intravenous (IV) treatment. This disparity may be attributable to data limitations.

## Discussion

This population-based propensity score matching study demonstrates a statistically significant reduction in the mortality rate among patients who received inhaled general anesthesia maintenance compared to propofol-based IV anesthetics in clinical stage III breast cancer. The results remained unchanged even after adjusting for age, gender, comorbidity, and medications. The sensitivity analysis further confirmed the robustness of our findings.

The literature on the impact of anesthetic techniques on advanced cancer stages is scant; nonetheless, our research indirectly suggests that the influence of anesthetic agents on breast cancer may differ across the disease's various stages. The immune system's capability to counter tumor progression involves the identification and elimination of cancer cells with mutational changes [24]. According to the established model, breast cancer immunoediting consists of three phases: elimination, equilibrium, and escape, each characterized by distinct immunological responses [25–27]. Anesthetic techniques may have different interactions with the immune system throughout these immunoediting phases, which warrants further

**Table 2. Comparison of overall mortality rate in stage III breast cancer patients considering different baseline characteristics, after adjusting for age, sex, comorbidities, and medications before propensity score matching.**

| Variable | Non-IA | | | IA | | | Univariate | | Multivariate | |
|---|---|---|---|---|---|---|---|---|---|---|
| | Event | Person-Year | IR | Event | Person-Year | IR | HR (95% CI) | P-value | HR (95% CI) | P-value |
| All | 390 | 6107 | 63.9 | 2141 | 48901 | 43.8 | 0.68(0.61, 0.76) | <0.001 | 0.82(0.72, 0.93) | 0.002 |
| Age, year | | | | | | | | | | |
| 20–49 | 72 | 1866 | 38.6 | 546 | 17258 | 31.6 | 0.82(0.64, 1.05) | 0.11 | 0.94(0.71, 1.26) | 0.69 |
| 50–64 | 149 | 2673 | 55.7 | 925 | 22784 | 40.6 | 0.72(0.61, 0.86) | <0.001 | 0.78(0.64, 0.95) | 0.02 |
| >64 | 169 | 1568 | 107.8 | 670 | 8859 | 75.6 | 0.69(0.58, 0.82) | <0.001 | 0.82(0.68, 0.99) | 0.04 |
| CCI | | | | | | | | | | |
| 0 | 133 | 2971 | 44.8 | 1531 | 38259 | 40.0 | 0.89(0.75, 1.07) | 0.21 | 0.90(0.74, 1.09) | 0.26 |
| 1–3 | 133 | 1919 | 69.3 | 435 | 8364 | 52.0 | 0.75(0.61, 0.91) | 0.003 | 0.76(0.61, 0.95) | 0.02 |
| >3 | 124 | 1217 | 101.9 | 175 | 2278 | 76.8 | 0.75(0.60, 0.94) | 0.01 | 0.76(0.58, 0.99) | 0.04 |
| BMI | | | | | | | | | | |
| <18 | 77 | 798 | 96.5 | 581 | 11968 | 48.6 | 0.52(0.41, 0.66) | <0.001 | 0.91(0.43, 1.92) | 0.81 |
| 18–24 | 122 | 2017 | 60.5 | 710 | 17480 | 40.6 | 0.68(0.56, 0.82) | <0.001 | 0.76(0.63, 0.93) | 0.01 |
| >24 | 191 | 3292 | 58.0 | 850 | 19453 | 43.7 | 0.74(0.63, 0.86) | <0.001 | 0.85(0.72, 1.00) | 0.49 |
| pStage | | | | | | | | | | |
| IIIA | 116 | 2898 | 40.0 | 692 | 22389 | 30.9 | 0.76(0.62, 0.93) | 0.007 | 0.93(0.75, 1.16) | 0.54 |
| IIIB | 24 | 327 | 73.5 | 205 | 3321 | 61.7 | 0.84(0.55, 1.28) | 0.40 | 1.20(0.73, 1.98) | 0.47 |
| IIIC | 137 | 1631 | 84.0 | 802 | 12548 | 63.9 | 0.76(0.63, 0.91) | 0.003 | 0.78(0.64, 0.96) | 0.02 |
| Unknown | 113 | 1251 | 90.3 | 442 | 10643 | 41.5 | 0.47(0.38, 0.58) | <0.001 | 0.72(0.56, 0.93) | 0.01 |
| Histology | | | | | | | | | | |
| IDC | 326 | 5200 | 62.7 | 1824 | 42431 | 43.0 | 0.68(0.61, 0.77) | <0.001 | 0.83(0.72, 0.95) | 0.005 |
| ILC | 27 | 427 | 63.2 | 120 | 2560 | 46.9 | 0.73(0.48, 1.12) | 0.15 | 0.95(0.59, 1.54) | 0.83 |
| Others | 37 | 480 | 77.0 | 197 | 3911 | 50.4 | 0.67(0.47, 0.95) | 0.03 | 0.73(0.48, 1.11) | 0.15 |
| Grade | | | | | | | | | | |
| Gr.1 | 18 | 353 | 51.0 | 78 | 3080 | 25.3 | 0.49(0.29, 0.82) | 0.006 | 0.73(0.36, 1.46) | 0.37 |
| Gr.2 | 144 | 3028 | 47.6 | 861 | 23310 | 36.9 | 0.75(0.63, 0.90) | 0.002 | 0.94(0.76, 1.15) | 0.53 |
| Gr.3 | 178 | 2213 | 80.4 | 979 | 18218 | 53.7 | 0.68(0.58, 0.80) | <0.001 | 0.81(0.67, 0.97) | 0.02 |
| Unknown | 50 | 513 | 97.5 | 222 | 4290 | 51.7 | 0.52(0.39, 0.71) | <0.001 | 0.63(0.44, 0.90) | 0.01 |
| LVSI | | | | | | | | | | |
| Negative | 202 | 3319 | 60.9 | 956 | 20754 | 46.1 | 0.85(0.65, 1.12) | 0.25 | 1.07(0.79, 1.44) | 0.67 |
| Positive | 127 | 1252 | 101.5 | 820 | 17353 | 47.3 | 0.75(0.64, 0.87) | <0.001 | 0.84(0.72, 0.99) | 0.04 |
| Unknown | 61 | 1536 | 39.7 | 365 | 10795 | 33.8 | 0.47(0.39, 0.57) | <0.001 | 0.66(0.51, 0.86) | 0.002 |
| Subtype | | | | | | | | | | |
| Luminal | 189 | 4052 | 46.7 | 976 | 27669 | 35.3 | 0.73(0.62, 0.85) | <0.001 | 0.86(0.73, 1.02) | 0.09 |
| HER2 enrich | 58 | 731 | 79.4 | 238 | 5121 | 46.5 | 0.59(0.44, 0.79) | <0.001 | 0.70(0.51, 0.96) | 0.03 |
| Basal | 58 | 483 | 120.2 | 328 | 3304 | 99.3 | 0.88(0.67, 1.17) | 0.38 | 1.03(0.76, 1.39) | 0.86 |
| Unknown | 85 | 842 | 101 | 599 | 12806 | 46.8 | 0.47(0.37, 0.59) | <0.001 | 0.56(0.39, 0.80) | 0.002 |
| Type of surgery | | | | | | | | | | |
| BCS | 269 | 4015 | 67 | 284 | 9490 | 29.9 | 1.07(0.74, 1.54) | 0.72 | 1.04(0.70, 1.54) | 0.86 |
| Mastectomy | 89 | 909 | 97.9 | 1582 | 34595 | 45.7 | 0.68(0.60, 0.77) | <0.001 | 0.83(0.71, 0.96) | 0.01 |
| Unknown | 32 | 1183 | 27.1 | 275 | 4816 | 57.1 | 0.61(0.48, 0.78) | <0.001 | 0.79(0.58, 1.08) | 0.15 |

[1] Abbreviation: HR, hazard ratio; CI, confidence interval; IR, incidence rate, also indicating mortality rate, per 1000 person-years; IV, intravenous anesthetic group; IA, inhaled anesthetic group; BMI, body mass index; CCI, Charlson Comorbidity Index; IDC, invasive ductal carcinoma; ILC, invasive lobular carcinoma; LVSI, lymphvascular space invasion; HER2, human epidermal growth factor receptor 2

[2] Adjusted HR: adjusted for age, sex, comorbidities and medications in Cox proportional hazards regression

**Table 3. Comparisons of mortality and recurrence rate between stage III breast cancer patients after propensity score matching.**

| | Non-IA (N = 1267) | | | IA (N = 1267) | | | Univariate | | Multivariate | |
|---|---|---|---|---|---|---|---|---|---|---|
| | Event | Person-Year | IR | Event | Person-Year | IR | HR (95% CI) | P-value | HR (95% CI) | P-value |
| Recurrence | 153 | 4810 | 31.8 | 217 | 6731 | 32.2 | 1.26(1.02, 1.55) | 0.03 | 1.28(1.04, 1.58) | 0.02 |
| Mortality | 295 | 5093 | 57.9 | 341 | 7226 | 47.2 | 0.83(0.71, 0.97) | 0.02 | 0.83(0.71, 0.98) | 0.02 |

Abbreviation: HR, hazard ratio; CI, confidence interval; IR, incidence rate, also indicating mortality rate, per 1000 person-years; IV, intravenous anesthetic group; IA, inhaled anesthetic group; Adjusted HR: adjusted for age, sex, comorbidities and medications in Cox proportional hazards regression.

investigation to tailor anesthetic strategies that consider the cancer's immunological profile at each stage.

Research has addressed how volatile anesthetics can modulate cancer signaling pathways [8, 28–30]. For example, anesthetics can impact cancer cell migration and invasion by modulating miRNA and MMP activity, which are pivotal to EMT processes [8, 28, 29, 31, 32]. Wu et al. have shown that Sevoflurane suppresses EMT in breast cancer cells by regulating miR-139–5p/ARF6, while Liu et al. demonstrated that a standard clinical concentration of Sevoflurane inhibits breast cancer cell proliferation through the upregulation of microRNA-203 [8, 30]. These findings align with our results, suggesting a potential protective effect of volatile anesthetics on overall survival in breast cancer patients.

Contrary to our findings, several meta-analyses and randomized controlled trials have shown no significant effects of anesthetics on recurrence-free survival (RFS) and overall survival (OS) in breast cancer, which included a mix of early-stage and advanced-stage patients [25, 33, 34]. Considering that the 5-year overall survival rates for various stages of breast cancer differ substantially, it is plausible that the duration of follow-up in previous studies may not have been adequate to discern long-term effects [35]. Our results, highlighting a lower overall mortality rate in patients receiving inhaled anesthesia, suggest a stage-specific protective effect that merits further investigation.

Moreover, our study indicates a particularly favorable survival outcome for patients over 50 with stage III breast cancer undergoing inhalation anesthesia. Aging is associated with reduced acute inflammatory responses, likely due to immunosenescence and changes in cytokine profiles [36]. Inhalation anesthetics are known to attenuate perioperative inflammatory responses, potentially through the modulation of inflammatory pathways [37]. The combination of age-related and anesthesia-induced reductions in inflammation may synergistically hinder oncogenic progression and lessen postoperative complications. Given the established role of persistent inflammation in tumor initiation, development, and metastasis, mitigating such inflammation could theoretically slow tumor growth and metastatic spread, thereby improving survival rates [38].

While these findings are preliminary, they highlight the importance of considering patient age and inflammatory status in anesthetic planning for oncologic surgery. It is crucial to further investigate these observations to understand the underlying mechanisms and to explore the strategic use of inhalation anesthetics to potentially improve cancer-related outcomes.

In our analysis, we observed an intriguing paradox: the IA group exhibited a higher overall recurrence rate yet a lower overall mortality rate. We attribute this phenomenon to varying data accuracy levels. The data accuracy for the overall mortality rate is high, but that for the overall recurrence rate is low. This discrepancy is due to the constraints imposed by the Taiwan Cancer Registry, which does not mandate institutions to report recurrence data. Although institutions may log any recurrence events occurring after the initial diagnosis, they typically

**(a)**

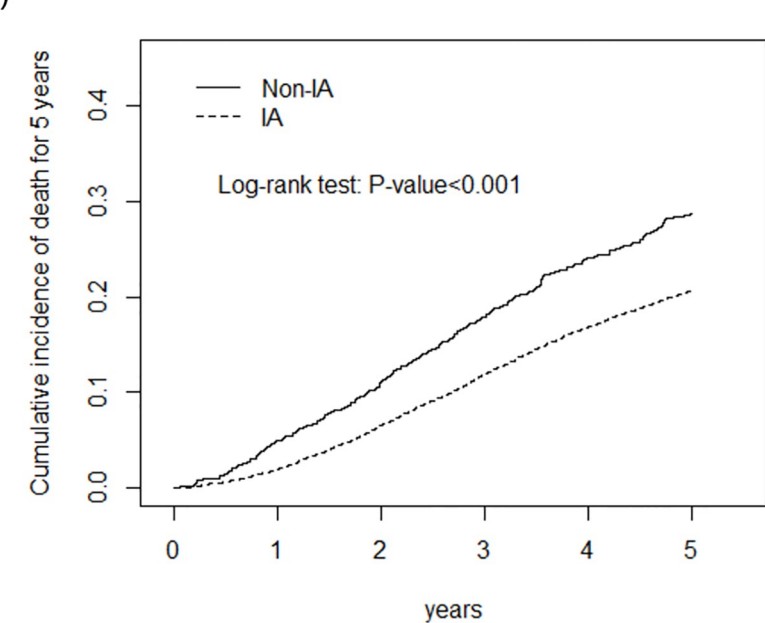

**(b)**

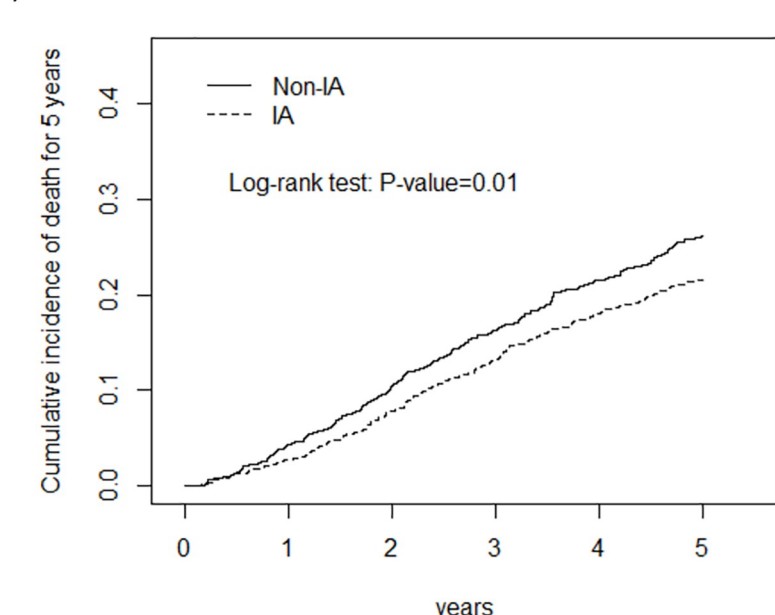

**Fig 3. The cumulative mortality rate among stage III breast cancer patients is lower with inhaled anesthetics (IA) compared to non-inhaled anesthetics (non-IA).** (A) Before propensity score matching; (B) After propensity score matching.

do so within 1 year of the diagnosis, which may be too short a period to document all recurrence events adequately. Moreover, our dataset lacked the comprehensive granularity commonly seen in clinical trials; for instance, we do not have data on progression-free survival (PFS) rates. Consequently, we chose to use the overall recurrence rate as the secondary

endpoint. Moreover, we can't have 3-year and 5-year recurrence rates as secondary endpoints since data logging is usually confined to the first year after diagnosis. In contrast, our mortality data, obtained from national death records in the NHIRD, are both accurate and reliable. Given these considerations, we decided to use the overall mortality rate as the primary endpoint. Further studies are needed to validate secondary endpoints, such as progression-free survival and recurrence-free survival, to ensure their accuracy and reliability. The secondary endpoint results from this study are only for reference.

This study possesses inherent limitations characteristic of retrospective designs. Our reliance on a medical claims database posed challenges, as such databases often lack the granularity required to detail specifics such as drug dosages, duration of anesthesia, and the precise choice of volatile anesthetics. This is a known limitation of many claims databases, where the primary purpose is billing rather than clinical documentation. Consequently, the data may not fully reflect the clinical scenario, carrying a risk of misclassification or omission of pertinent clinical details. Additionally, it must be acknowledged that stage III breast cancer encompasses sub-stages IIIA, IIIB, and IIIC, each with its own inherent heterogeneity. Due to data constraints and the limitations of the statistical methodologies available, separate analyses of each subcategory were not feasible. As a result, these subcategories were aggregated for analysis. This pooling approach may have resulted in the loss of detailed information relevant to the nuances of each subcategory, potentially obscuring specific trends and outcomes associated with each distinct subgroup. Given these challenges and limitations, it is evident that future investigations should consider prospective, randomized controlled trials (RCTs). An RCT would provide a more controlled environment to meticulously examine these variables, ensuring superior data accuracy and clinical relevance.

## Conclusion

In conclusion, our study provides compelling evidence that stage III breast cancer patients who received inhaled anesthetics experienced significantly lower overall mortality rates compared to those in the intravenous propofol-based maintenance group. Specifically, patients over the age of 50 who underwent surgery with propofol-based anesthesia maintenance showed a correlation with a reduced mortality rate. These findings highlight the potential impact of anesthesia technique on patient outcomes in breast cancer. However, further clinical investigations are necessary to validate and expand upon these results. This research has the potential to inform and improve treatment strategies for breast cancer patients, ultimately contributing to better patient care and outcomes in the future.

## Supporting information

**S1 Fig.**
(TIFF)

**S2 Fig.**
(TIFF)

**S3 Fig.**
(TIFF)

**S4 Fig.**
(ZIP)

## Acknowledgments

We are grateful to Health Data Science Center, China Medical University Hospital for providing administrative and technical support. Their invaluable contributions have greatly facilitated the execution and completion of this study.

## Author Contributions

**Conceptualization:** Chin Kuo.

**Data curation:** Cheng-Li Lin.

**Formal analysis:** Emily Tzu-Jung Kuo, Cheng-Li Lin.

**Investigation:** Emily Tzu-Jung Kuo.

**Methodology:** Cheng-Li Lin.

**Software:** Cheng-Li Lin.

**Supervision:** Chin Kuo.

**Validation:** Chin Kuo.

**Writing – original draft:** Emily Tzu-Jung Kuo, Chin Kuo.

**Writing – review & editing:** Emily Tzu-Jung Kuo, Chin Kuo.

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
