## [Decision Letter · Decision Letter 0]

7 Sep 2023

PONE-D-23-22217Inhaled Anesthesia Associated with Reduced Mortality in Patients with Stage III Breast Cancer: A Population-Based StudyPLOS ONE

Dear Dr. Kuo,

Thank you for submitting your manuscript to PLOS ONE. After careful consideration, we feel that it has merit but does not fully meet PLOS ONE’s publication criteria as it currently stands. Therefore, we invite you to submit a revised version of the manuscript that addresses the points raised during the review process.

We look forward to receiving your revised manuscript.

Kind regards,

Alok K Mishra

Academic Editor

PLOS ONE

“We gratefully acknowledge the support of the China Medical University Hospital for 208

funding this study. Inaddition, we are grateful to Health Data Science Center, China 209

Medical University Hospital for providing administrative, technical and funding support. 210

Their invaluable contributions have greatly facilitated the execution and completion of 211

this study.”

“This study was sponsored by China Medical University Hospital. The funders had no role in study design, data collection and analysis, decision to publish, or preparation of the manuscript.”

Reviewers' comments:

Reviewer's Responses to Questions

**Comments to the Author**

1. Is the manuscript technically sound, and do the data support the conclusions?

Reviewer #1: Yes

Reviewer #2: Yes

Reviewer #3: Yes

2. Has the statistical analysis been performed appropriately and rigorously? 

Reviewer #1: Yes

Reviewer #2: No

Reviewer #3: Yes

3. Have the authors made all data underlying the findings in their manuscript fully available?

Reviewer #1: Yes

Reviewer #2: Yes

Reviewer #3: No

4. Is the manuscript presented in an intelligible fashion and written in standard English?

Reviewer #1: Yes

Reviewer #2: Yes

Reviewer #3: Yes

5. Review Comments to the Author

Reviewer #1: The impact of propofol-based intravenous anesthetic maintenance versus inhaled anesthetic maintenance on the mortality of Stage III Breast Cancer was examined in this comprehensive analysis. This investigation utilized a sizable population-based cohort to shed light on the controversial role of different anesthetics in the prognosis of oncology patients. The outcomes of this research contribute substantial evidence to this critical domain. The study exhibited meticulous design, credible outcomes, and thorough documentation of methodologies.

1.In addition to assessing overall survival (OS) as the primary outcome, could we also explore secondary endpoints? Furthermore, is there potential to capture progression-free survival (PFS) rates?

2.It is advisable to also address potential confounding variables through multifactorial COX regression analyses. Considering an unmatched total sample would help us gauge the consistency of outcomes across different parameters.

3. Incorporating subgroup analyses could enhance the comprehensiveness of the study, offering insights into specific demographic or clinical factors that might influence the observed effects.

4.The section outlining limitations could benefit from expansion, detailing potential constraints encountered during the study. Additionally, highlighting the strengths of this investigation would provide a more holistic perspective.

Reviewer #2: Overall impression

This is an important study to publish and discuss.

Specific comments

The strength is that the data from Taiwan’s National Health Insurance Research Database and Taiwan Cancer Registry probably are of good quality, as far as can be judged from abroad - definitely, the amount of data is impressive.

The background information on possible mechanisms behind the finding is satisfactory, although we have many more mechanisms to digest regarding the possible "good" properties of propofol in terms of a possible beneficial effect in cancer surgery. Today, no one can say with certainty which mechanism(s) are actually of clinical importance.

The weakness of the study is that the study is retrospective, and therefore, the result just gives us a hint of what might be the truth. So regardless of whether both the data quality and the amount of data are satisfactory, and that the statistical and epidemiological methodology is good with e.g. propensity score matching, the retrospective design, and not least important, the use of inhalation anaesthetics or propofol for maintenance of anaesthesia was of course not randomised (and the proportion of patients who received propofol was relatively small) means that the results must be taken with a large pinch of salt.

Another topic to comment on is the identification of stage III cancer as a specific entity. As far as I can tell, as a simple anaesthetist, the authors want us to believe that stage III cancer is well defined. The TNM system, found in Table 1, illustrates that stage III cancer is not quite homogeneous:

Stage IIIA breast cancer is the same as:

T0 N2 M0,

T1 N2 M0,

T2 N2 M0,

T3 N1 M0,

T3 N2 M0.

Stage IIIB is the same as:

T4 N0 M0,

T4 N1 M0,

T4 N2 M0.

Stage IIIC is the same as:

Any T N3 M0.

Although I have been doing part-time research for 30+ years, I am still a simple anaesthetist without advanced knowledge of statistics and epidemiology. So, you may excuse me for asking about the statistical adjustment of the propensity score matched cohort. Isn't adjusting an already "adjusted" cohort the same as over-adjusting?

If this manuscript is intended to reach clinicians, I am not satisfied with the method of presenting the difference between the cohorts. We clinicians are more likely to understand differences when they are expressed as five-year survival (or for that matter mortality), or even more interestingly in the case of breast cancer, ten-year survival.

To summarise, I would like to see:

* a more humble discussion considering the retrospective design,

* a discussion of the heterogeneity hidden in the concept of stage III cancer,

* a consideration of a more clinically appropriate presentation of the difference in mortality (or survival) between the groups.

Reviewer #3: Kuo et al. have indicated that inhaled anesthesia is associated with reduced mortality in patients with stage III breast cancer. Their research analysis with 10,896 breast cancer patients offers valuable insights for surgeons/investigators to select IA or IV for required surgery. The authors have pointed out that using IA might benefit patients over IV. This well-written research article makes a valuable contribution to the field by bringing attention to the surgeons that the selection of anesthesia is extremely important for patient survival. I propose a few major and minor suggestions for your consideration.

Major points:

1. This research article has only two figures. It would be helpful to readers if the author could mention the current global status of breast cancer by showing a figure. Other figures might include the different methods of Anesthesia used during breast cancer surgery.

2. What are the different factors that contribute to the selection of a particular method, such as IA and IV? Which method is mostly/generally preferred for breast cancer patients?

Minor points:

1. On page 1, two symbols about contributed equally and current address are mentioned, but these symbols are not present with the author's name.

2. Reference formatting needs to be more consistent. Citation references normally go inside the sentence. In several places, references are mentioned after the full stop.

3. Line 47 on page 3, incomplete sentence.

4. Line 142-143 on page 5, “Research studies…” is mentioned, but only one reference is cited.

6. PLOS authors have the option to publish the peer review history of their article (what does this mean?). If published, this will include your full peer review and any attached files.

Reviewer #1: No

Reviewer #2: No

Reviewer #3: No

---

## [Author Response · Author response to Decision Letter 0]

17 Dec 2023

Reviewer 1:

Q1.In addition to assessing overall survival (OS) as the primary outcome, could we also explore secondary endpoints? Furthermore, is there potential to capture progression-free survival (PFS) rates? 

Response: We fully recognize the importance of secondary endpoints, especially in the context of cancer prognosis. Regarding progression-free survival (PFS) data, our dataset from the Taiwan Cancer Registry does not have a designated section for consistently assessing disease progression at standardized time intervals using the RECIST criteria. This level of detail, which is typically a hallmark of clinical trial data, was notably absent in most entries in our dataset. Consequently, extracting progression-free survival from our existing records proved challenging. Given these data limitations, we opted for overall recurrence rate as a secondary endpoint. This choice is both pragmatic and pertinent for measuring disease progression within the context of our study. Nevertheless, we acknowledge that the quality of data regarding the overall recurrence rate may be suboptimal. Taiwanese cancer databases do not mandate periodic updates on recurrence data; thus, the data is not frequently refreshed beyond the first year post-diagnosis. Consequently, this information is often only available for reference. 

Actions: We have included the overall recurrence rate as a surrogate endpoint in the ’Outcome’ subsection of the ’Materials and Methods’ section, line 82-87. Additionally, an extended discussion is presented from lines 206 to 223 in the ’Discussion’ section. The revised portions have been highlighted in yellow. 

Q2. It is advisable to also address potential confounding variables through multifactorial COX regression analyses. Considering an unmatched total sample would help us gauge the consistency of outcomes across different parameters. 

Response: Thank you for your insightful comments, we have now included an unmatched total sample analysis in our analyses. 

Actions: The revised Table 2 presents the results for the total sample. 

Q3.Incorporating subgroup analyses could enhance the comprehensiveness of the study, offering insights into specific demographic or clinical factors that might influence the observed effects. 

Response: In response to your suggestion, we conducted subgroup analyses, paying special attention to discernible differences. A notable finding from this subgroup analysis indicates that age significantly influences outcomes. Specifically, the IA group demonstrated more favorable outcomes in individuals over 50. 

Actions: We have addressed the responses of different age groups in both the ’Results’ section, lines 132 to 134, and the ’Discussion’ section, lines 190 to 200. 

Q4.The section outlining limitations could benefit from expansion, detailing potential constraints encountered during the study. Additionally, highlighting the strengths of this investigation would provide a more holistic perspective. 

Response: Our study is inherently constrained by its retrospective design, which does not offer the precision and specificity characteristic of prospective studies tailored to address distinct questions. While our focus on overall survival (OS) provides a reliable metric, the inclusion of other outcomes, such as progression-free survival, would have enriched the oncological context of our research. Regrettably, our database did not house this specific data, precluding us from analyzing secondary endpoints. A notable strength of our investigation lies in leveraging a national database. Gathering extensive data on stage III breast cancer from a singular institution can be formidable, often spanning numerous years. The national database enabled us to obtain a comprehensive view of stage-specific breast cancers. While previous research has often amalgamated early and late-stage breast cancer responses to anesthesia, we specifically examined late-stage responses. Given existing literature that highlights differential immune reactions in early versus late cancer stages, our focused approach augments the understanding of anesthesia responses in advanced breast cancer. 

Actions: We have expanded the ’Limitations’ section in the ’Discussion’ from lines 224 to 240. 

Reviewer 2:

The strength is that the data from Taiwan’s National Health Insurance Research Database and Taiwan Cancer Registry probably are of good quality, as far as can be judged from abroad - definitely, the amount of data is impressive. The background information on possible mechanisms behind the finding is satisfactory, although we have many more mechanisms to digest regarding the possible ”good” properties of propofol in terms of a possible beneficial effect in cancer surgery. Today, no one can say with certainty which mechanism(s) are actually of clinical importance. The weakness of the study is that the study is retrospective, and therefore, the result just gives us a hint of what might be the truth. So regardless of whether both the data quality and the amount of data are satisfactory, and that the statistical and epidemiological methodology is good with e.g. propensity score matching, the retrospective design, and not least important, the use of inhalation anaesthetics or propofol for maintenance of anaesthesia was of course not randomised (and the proportion of patients who received propofol was relatively small) means that the results must be taken with a large pinch of salt. 

Q1. Another topic to comment on is the identification of stage III cancer as a specific entity. As far as I can tell, as a simple anaesthetist, the authors want us to believe that stage III cancer is well defined. The TNM system, found in Table 1, illustrates that stage III cancer is not quite homogeneous: 

Stage IIIA breast cancer is the same as: T0 N2 M0,

T1 N2 M0,

T2 N2 M0, 

T3 N1 M0,

T3 N2 M0.

Stage IIIB is the same as:

T4 N0 M0,

T4 N1 M0,

T4 N2 M0.

Stage IIIC is the same as:

Any T N3 M0.

Response: The reviewer’s observation regarding the subdivision of stage 3 breast cancer into categories 3A, 3B, and 3C is indeed well-founded. Each category can be further delineated into distinct classifications based on the tumor size and nodal involvement (TN categories). If one were to parse out each TN category for detailed analysis, the resulting subpopulations would be exceedingly small, presenting considerable statistical challenges due to limited sample sizes which may not yield statistically significant results. Therefore, in our study, we have adopted an approach in Table 1 that focuses on broader trends by initially comparing outcomes within the entirety of stage 3 patients. Specifically, we have directed our analysis towards discerning the differences between patients with pronounced tumor size (T category) and those with extensive nodal involvement (N category). This strategy allows us to observe overarching patterns and potential prognostic differences in survival and outcomes between these two subsets of patients with advanced disease. This methodological decision enables us to maintain statistical robustness while still providing insightful distinctions within a broadly defined patient cohort. 

Q2. Although I have been doing part-time research for 30+ years, I am still a simple anaesthetist without advanced knowledge of statistics and epidemiology. So, you may excuse me for asking about the statistical adjustment of the propensity score matched cohort. Isn’t adjusting an already ”adjusted” cohort the same as over-adjusting? 

Response: Thank you for your insightful comments regarding the application of propensity-score matching (PSM) and the subsequent use of multivariate analysis in our study. Under ideal circumstances, PSM would indeed obviate the need for further multivariate analysis, as it aims to simulate a randomized controlled trial by creating a balanced cohort in which the treatment and control groups are matched on confounding variables. In such a scenario, univariate and multivariate analyses would, theoretically, yield consistent results. However, the inherent complexity of real-world data often precludes the achievement of perfect balance between groups, even after meticulous propensity score matching. Subtle imbalances may persist due to unmeasured confounders or the limited overlap in the distribution of propensity scores, which can result in residual confounding. The decision to perform additional multivariate analysis post-PSM in our study is a deliberate one, intended to enhance the robustness of our analytical approach. By incorporating multivariate analysis, we aim to control for any remaining imbalances and confirm the consistency of the PSM results. This step is particularly crucial given the assumption of the PSM that all confounders are measured and correctly included in the model. The multivariate analysis provides an additional layer of adjustment, mitigating the impact of any potential unobserved heterogeneity. Furthermore, multivariate analysis allows us to assess the effect of each covariate on the outcome, controlling for other variables, which is especially important when the propensity score model may not fully account for the complexities of the data. This approach acknowledges the propensity of real-world data to defy the strict assumptions of statistical models and seeks to solidify the conclusions drawn from the observed associations. We believe that this complementary use of PSM followed by multivariate analysis does not reflect a lack of confidence in the matching process but rather a prudent acknowledgment of the limitations inherent in observational data. It is a strategy designed to ensure that our findings are not only statistically sound but also as close to the underlying biological reality as possible. 

If this manuscript is intended to reach clinicians, I am not satisfied with the method of presenting the difference between the cohorts. We clinicians are more likely to understand differences when they are expressed as five-year survival (or for that matter mortality), or even more interestingly in the case of breast cancer, ten-year survival. 

To summarise, I would like to see:

Q3. A more humble discussion considering the retrospective design. 

Actions: We have expanded the ’Limitations’ section within the ’Discussion’ to encompass lines 224 to 230, with a particular emphasis on the limitations inherent in retrospective studies. 

Q4. A discussion of the heterogeneity hidden in the concept of stage III cancer. 

Actions: We have extended the ’Limitations’ section in the ’Discussion’ to cover lines 230 to 237, with specific focus on the inherent heterogeneity within the classification of stage III cancer. 

Q5. A consideration of a more clinically appropriate presentation of the difference in mortality (or survival) between the groups. 

Actions: Thanks for great suggestion! We have included both 3-year and 5-year mortality rates, as these metrics are more readily comprehensible to physicians. 

Reviewer 3:

Major points:

Q1. This research article has only two figures. It would be helpful to readers if the author could mention the current global status of breast cancer by showing a figure. Other figures might include the different methods of Anesthesia used during breast cancer surgery. 

Actions: We have included Fig 1 to illustrate the global status of breast cancer. Additionally, Fig 2 has been added to provide an overview of the topic addressed in this manuscript. 

Q2. What are the different factors that contribute to the selection of a particular method, such as IA and IV? Which method is mostly/generally preferred for breast cancer patients? 

Response: Breast cancer surgeries employ various anesthetic techniques, including regional anesthesia, IV sedation, general anesthesia, and combinations thereof. Stage III breast cancer surgeries, which often encompass complex procedures like Modified Radical Mastectomy(MRM), breast-conserving surgery(BCS) with sentinel lymph node biopsy(SLND), BCS with lymph node dissection, or reconstruction, primarily utilize general anesthesia. These procedures are typically lengthier than those for stages I or II, where BCS+SLND is more common. Research suggests that anesthetic choice doesn’t significantly influence post-surgery breast pain. Therefore, the selection between regional and general anesthesia is often guided by clinicians’ preferences, with the latter being the prevalent choice. Within general anesthesia, two primary techniques are used: inhalational and total intravenous anesthesia (TIVA). Both techniques have long-standing safety records. The choice is often dictated by specific patient-related factors, such as the risk of post-operative nausea and vomiting or the uncommon risk of malignant hyperthermia. It’s pertinent to note the ongoing debate regarding the influence of anesthetics on cancer and immune responses. In the backdrop of Taiwan’s healthcare system, the introduction of Processed EEG Monitoring in 2007 marked a shift in favor of TIVA. However, TIVA lacks the minimal alveolar concentration (MAC) that was traditionally employed for depth monitoring and is generally costlier than inhalational methods. Additionally, Processed EEG Monitoring remains an out-of-pocket expense for breast cancer surgery in Taiwan. Considering the heightened risk of post-operative nausea and vomiting among Asian breast cancer patients, propofol-based general anesthesia has gained traction. However, its impact on cancer outcomes is still a subject of debate, which underpins the essence of our study. 

Minor points:

Q1. On page 1, two symbols about contributed equally and current address are mentioned, but these symbols are not present with the author’s name. 

Response: We apologize for the oversight regarding the symbols on page 1. This was an inadvertent error arising from the use of a LaTeX template. Thank you for bringing this to our attention. 

Actions: We removed the unnecessary symbols. 

Q2. Reference formatting needs to be more consistent. Citation references normally go inside the sentence. In several places, references are mentioned after the full stop. 

Response: Thank you for addressing this issue. We reviewed the entire manuscript and ensured all citations were consistently placed inside the sentences. 

Actions: The corrected formatting has been applied, and we have double-checked to confirm the consistency of reference formatting. 

Q3. Line 47 on page 3, incomplete sentence.

Response: We have since revised it, ensuring it is a complete and coherent sentence. 

Actions: We have corrected it and highlighted the revised portion at line 51 for easier reference. 

Q4. Line 142-143 on page 5, “Research studies. . . ” is mentioned, but only one reference is cited. 

Response: Thank you for pointing out the oversight in our citation. We have corrected this by adding multiple relevant references to support the statement. 

Actions: We have addressed this by adding the relevant references at line 174. The line number has changed due to revisions in the manuscript.

---

## [Decision Letter · Decision Letter 1]

11 Jan 2024

Inhaled Anesthesia Associated with Reduced Mortality in Patients with Stage III Breast Cancer: A Population-Based Study

PONE-D-23-22217R1

Dear Dr. Kuo,

We’re pleased to inform you that your manuscript has been judged scientifically suitable for publication and will be formally accepted for publication once it meets all outstanding technical requirements.

Kind regards,

Alok K Mishra

Academic Editor

PLOS ONE

Reviewers' comments:

Reviewer's Responses to Questions

**Comments to the Author**

1. If the authors have adequately addressed your comments raised in a previous round of review and you feel that this manuscript is now acceptable for publication, you may indicate that here to bypass the “Comments to the Author” section, enter your conflict of interest statement in the “Confidential to Editor” section, and submit your "Accept" recommendation.

Reviewer #1: All comments have been addressed

Reviewer #3: All comments have been addressed

2. Is the manuscript technically sound, and do the data support the conclusions?

Reviewer #1: Yes

Reviewer #3: Yes

3. Has the statistical analysis been performed appropriately and rigorously? 

Reviewer #1: Yes

Reviewer #3: Yes

4. Have the authors made all data underlying the findings in their manuscript fully available?

Reviewer #1: Yes

Reviewer #3: No

5. Is the manuscript presented in an intelligible fashion and written in standard English?

Reviewer #1: Yes

Reviewer #3: Yes

6. Review Comments to the Author

Reviewer #1: The author answered my question very well and this article is suitable to be published in its present form

Reviewer #3: This revision has significantly improved the manuscript. The authors highlighted the manuscript's changes and effectively explained the detailed responses to the reviewer’s comments. I would recommend the manuscript for publication.

7. PLOS authors have the option to publish the peer review history of their article (what does this mean?). If published, this will include your full peer review and any attached files.

Reviewer #1: No

Reviewer #3: No

---

## [Editor Report · Acceptance letter]

21 Feb 2024

PONE-D-23-22217R1 

PLOS ONE

Dear Dr. Kuo, 

I'm pleased to inform you that your manuscript has been deemed suitable for publication in PLOS ONE. Congratulations! Your manuscript is now being handed over to our production team.

Kind regards, 

on behalf of

Dr. Alok K Mishra 

Academic Editor

PLOS ONE